# Genome-Wide Identification and Expression Analysis of *CAMTA* Genes in Cassava Under Abiotic Stresses

**DOI:** 10.3390/plants14243743

**Published:** 2025-12-08

**Authors:** Feilong Yu, Chenyu Lin, Xianhai Xie, Xiaohui Yu, Xin Guo

**Affiliations:** School of Tropical Agriculture and Forestry, Hainan University, Haikou 570228, China; yu1484754611@163.com (F.Y.); xianhaixie0901@163.com (X.X.)

**Keywords:** cassava, *CAMTA* gene family, abiotic stress, gene expression

## Abstract

Cassava (*Manihot esculenta* Crantz) is a major dual-purpose crop in tropical and subtropical regions, but its growth and yield are significantly constrained by abiotic stresses. Calmodulin-binding transcription activators (CAMTAs) are key regulators involved in plant development and stress responses. In this research, six *CAMTA* genes (*MeCAMTAs*) were identified from the first telomere-to-telomere (T2T) genome assembly of cassava, and these genes are distributed on four chromosomes. These genes are divided into three different subfamilies based on phylogenetic relationships. Homology analysis shows that there is one pair of replication gene pairs. Analysis of cis-acting elements reveals that the promoter regions of the *MeCAMTA* gene family contain cis-acting elements responsive to hormones, abiotic stresses, growth and development, and light. Through analysis of the expression patterns of *MeCAMTAs* in different tissues, *MeCAMTAs* are expressed in all tissues, among which *MeCAMTA3* and *MeCAMTA4.1* are mainly expressed in stems, while *MeCAMTA1*, *MeCAMTA2*, *MeCAMTA4.2*, and *MeCAMTA6* are mainly expressed in roots. qRT-PCR analysis shows that *MeCAMTAs* exhibit dynamic expression patterns under different abiotic stress treatments. Therefore, this study provides a certain reference basis for the research on the abiotic stress response mechanism of cassava and also provides potential genetic resources for the stress-resistant breeding of cassava.

## 1. Introduction

Cassava is a major tuber crop cultivated extensively in tropical and subtropical regions, serving as a staple food for over 800 million people and ranking as the fourth-largest source of calories worldwide [1,2,3]. Its exceptional adaptability to marginal soils and drought, together with high starch content, underpins its importance not only as a food crop but also as a raw material for pharmaceuticals, biomaterials, animal feed, biofuels, and ethylene production [4,5]. However, abiotic stresses such as drought and salinity severely impair cassava growth, development, and yield, posing significant threats to food security [6]. Enhancing stress resilience is therefore essential for sustainable agricultural productivity, as crops with poor stress tolerance often require excessive water and fertilizer inputs, increasing environmental pressure [7].

Calcium ions (Ca^2+^) act as ubiquitous secondary messengers in plants, mediating a wide range of signaling pathways in growth, development, and stress responses [8]. Numerous studies have confirmed that Ca^2+^, as one of the key messengers, plays a crucial role in regulating plant growth and development, as well as responding to abiotic stresses [9]. In plant cells, free Ca^2+^ responds to various stimuli and transmits signals to be recognized and decoded by Ca^2+^ sensors. Studies have shown that Ca^2+^ sensors undergo conformational changes upon binding with Ca^2+^, thereby regulating their own activity, while also interacting with other proteins and modulating protein functions. Ca^2+^ sensors primarily include calmodulin (CaM), CaM-like proteins (CML), calcineurin B-like proteins (CBL), and calcium-dependent protein kinase (CDPK) [10,11,12]. Among these, CaM is a highly conserved Ca^2+^ sensor in eukaryotes [13,14], which interacts with diverse CaM-binding proteins (CBPs) such as transcription factors, phosphatases, ion channels, and metabolic enzymes in a Ca^2+^-dependent manner [9,13,14,15,16,17]. In plants, several transcription factor families—including calmodulin-binding transcription activators (CAMTAs), WRKY, and CBP60—contain CBPs that play pivotal roles in stress signaling [18].

The CAMTA family represents key Ca^2+^/CaM-regulated transcription factors involved in plant responses to both abiotic and biotic stresses [19]. CAMTA plays an indispensable role in plant responses to abiotic stresses such as drought, salinity, and cold stress, as well as biotic stresses. *CAMTA1*, *CAMTA2*, and *CAMTA3* synergistically activate *CBF1*, *CBF2*, and *CBF3* under low-temperature conditions, thereby improving cold tolerance in *Arabidopsis thaliana* [20]. AtCAMTA1 modulates plant drought responses by regulating the expression of multiple stress-responsive genes, such as *RAB18*, *COR78*, *CBF1*, and *ERD7* [21]. Overexpression of *CAMTA* genes enhances peroxidase activity, improving salinity tolerance in chickpeas [22]. In *atcamta3* mutant lines, salicylic acid (SA) accumulation is elevated, which enhances disease resistance in *Arabidopsis thaliana* [23]. These studies indicate that *CAMTA* genes play crucial roles in plants by regulating responses to both biotic and abiotic stresses.

Although cassava is of great economic and agronomic importance, a comprehensive characterization of the *CAMTA* gene family in this crop is still lacking, and its possible functions in abiotic stress adaptation have yet to be thoroughly investigated. In this study, we identified six *CAMTA* genes from the recently released telomere-to-telomere (T2T) genome assembly of cassava. We performed comprehensive analyses of their phylogeny, chromosomal distribution, gene structure, conserved motifs, cis-regulatory elements, and predicted protein–protein interaction networks. Furthermore, quantitative real-time PCR (qRT-PCR) was used to assess their tissue-specific expression and transcriptional responses to multiple abiotic stresses. These results provide a foundation for elucidating CAMTA-mediated regulatory mechanisms in cassava and offer valuable genetic resources for breeding stress-tolerant varieties.

## 2. Results

### 2.1. Identification and Basic Characteristics of MeCAMTA Genes

A total of six *CAMTA* genes were identified through a genome-wide search of the cassava T2T genome. Based on sequence homology with *Arabidopsis thaliana*, they were designated as *MeCAMTA1* to *MeCAMTA6* (Table 1 and Figure 1). Analysis of their physicochemical properties showed that the encoded proteins ranged from 925 to 1079 amino acids in length, with molecular weights between 104.76 and 120.79 kDa. The predicted isoelectric points (pI) varied from 5.49 to 6.99. All six proteins exhibited negative GRAVY values, indicating hydrophilic characteristics. Among them, only *MeCAMTA6* had an instability index below 40, suggesting protein stability, while the remaining five members showed values above 40, implying relative instability. Subcellular localization prediction further revealed that all six *MeCAMTAs* are localized in the nucleus (Table 1).

### 2.2. Phylogenetic Construction of the MeCAMTA Gene Family

To explore the evolutionary relationships of the *CAMTA* gene family in cassava, a phylogenetic tree was constructed using homologous protein sequences from *Arabidopsis thaliana* and *Oryza sativa* with the Neighbor-Joining (NJ) method (Figure 1). The results revealed that the six MeCAMTA members were classified into three subfamilies. Group I included *MeCAMTA1*, *MeCAMTA2*, and *MeCAMTA3*; Group III comprised *MeCAMTA4.1* and *MeCAMTA4.2*; only *MeCAMTA6* belongs to group IV. Notably, CAMTA proteins grouped within the same phylogenetic branch show high sequence homology, supporting the likelihood of conserved biological functions. In addition, the five species possess comparable numbers of CAMTA family members, indicating evolutionary conservation and implying that these genes likely perform essential biological roles.

### 2.3. Chromosomal Localization and Homology Analysis of MeCAMTAs

Chromosomal localization analysis revealed that *MeCAMTA* genes are unevenly distributed across four cassava chromosomes. Two chromosomes each harbor two genes: LG03 contains *MeCAMTA2* and *MeCAMTA6*, while LG12 contains *MeCAMTA3* and *MeCAMTA4.1*. The remaining two chromosomes each contain one gene, with *MeCAMTA1* located on LG6 and *MeCAMTA4.2* on LG13 (Figure 2A).

Homology analysis of the cassava T2T genome using MCScanX in TBtools v2.114 identified a pair of duplicated genes (*MeCAMTA4.1*/*MeCAMTA4.2*), suggesting a gene duplication event in cassava (Figure 2B). We analyzed gene duplication events using the DupGen_finder tool. The results showed that one pair of gene pairs originated from the whole-genome duplication (WGD) event (*MeCAMTA4.1*/*MeCAMTA4.2*), and two pairs of gene pairs originated from transposition duplication (TRD) (*MeCAMTA1*/*MeCAMTA3* and *MeCAMTA2*/*MeCAMTA3*). Furthermore, to further investigate the evolutionary selection pressure on duplicated genes, we calculated the nonsynonymous (Ka) and synonymous (Ks) substitution rates and determined the Ka/Ks ratio for this duplicated gene pair Appendix A. The Ka/Ks ratio was 0.27, which is below 1.0, indicating that this gene pair has undergone purifying selection during the evolutionary process. These results suggest that this gene pair is highly evolutionarily conserved within *MeCAMTA* and may possess important functional roles. To further investigate the evolutionary conservation of the *MeCAMTA* family, collinearity analysis was performed with dicotyledonous (*Arabidopsis thaliana*), monocotyledonous (*Oryza sativa*), and *Solanum tuberosum* species (Figure 2C and Appendix A). The results revealed five orthologous gene pairs between *Manihot esculenta* and *Arabidopsis thaliana*, seven between *Manihot esculenta* and *Oryza sativa*, and eight between *Manihot esculenta* and *Solanum tuberosum*. These findings reveal that the *CAMTA* genes in *Manihot esculenta*, *Arabidopsis thaliana*, *Solanum tuberosum*, and *Oryza sativa* exhibit high conservation and evolutionary homology.

### 2.4. Structural Features of the MeCAMTA Gene Family

To explore the structural characteristics of *MeCAMTA* genes, we analyzed their phylogenetic relationships (Figure 3A), conserved motifs, conserved domains, and gene structures using the MEME online tool and TBtools v2.114. The results showed that all *MeCAMTA* members contained 10 conserved motifs, designated as Motifs 1–10 (Figure 3B). In terms of conserved domains, except for *MeCAMTA2*, all other members contained four typical domains: CG-1, TIG, ANK (Ankyrin repeat), and IQ. Interestingly, *MeCAMTA2* harbored a unique functional domain belonging to the COG5022 superfamily (Figure 3C). Gene structure analysis revealed that the number of exons in *MeCAMTA* gene family members ranged from 12 to 13, while the number of introns varied between 11 to 12 (Figure 3D). These results indicate that *MeCAMTAs* are relatively conserved during evolution.

### 2.5. Analysis of Cis-Acting Elements in MeCAMTAs

Using the PlantCARE online tool to predict and analyze the promoter region 1500 bp upstream of the transcription start site of the *MeCAMTA* gene, we identified a total of 107 cis-acting elements in the promoter region of *MeCAMTAs* (Figure 4A). These mainly include abiotic stress-responsive elements (12), such as low temperature (LTR), anaerobic (ARE), and drought (CCAAT-box); hormone-responsive elements (32), such as SA (TCA-element), abscisic acid (ABA) (ABRE), methyl jasmonate (MeJA) (CGTCA-motif and TGACG-motif), gibberellin (GA) (GARE-motif), and auxin (IAA) (TGA-element); growth and development-responsive elements (6), such as meristem (CAT-box), endosperm (GCN4_motif), and storage (O2-site); and light-responsive elements (56), such as G-box, Box 4, etc. (Figure 4B). The results indicate that *MeCAMTAs* play important roles in physiological processes involved in plant growth and development, abiotic stress responses, and hormone responses.

### 2.6. Predicted Protein–Protein Interaction Network of MeCAMTAs

The interaction networks of MeCAMTA proteins were predicted using the STRING database (Figure 5). The results showed that MeCAMTA1 and MeCAMTA3 interacted with ANN1, Annexin, FAM192A, and COQ4; MeCAMTA2 and MeCAMTA6 both interacted with threonine protein kinase, calreticulin, FAM192A, and COQ4, while MeCAMTA4.1 and MeCAMTA4.2 interacted with threonine protein kinase, calreticulin, FAM192A, COQ4, and a C3H1-type domain protein. Overall, most of the interacting proteins are associated with Ca^2+^ signal transduction, suggesting that this interaction network provides important insights into the potential functions of MeCAMTAs.

### 2.7. Tissue-Specific Expression Profiles of MeCAMTA Genes

To investigate the tissue-specific expression patterns of *MeCAMTAs*, we examined their transcript levels in leaves, flowers, roots, seeds, fruits, stems, and root tubers (Figure 6). The results revealed distinct expression profiles among *MeCAMTA* family members. *MeCAMTA1*, *MeCAMTA2*, *MeCAMTA4.2*, and *MeCAMTA6* showed the highest expression in roots but the lowest in root tubers. In contrast, *MeCAMTA3* and *MeCAMTA4.1* exhibited peak expression in stems. These findings suggest that *MeCAMTA3* and *MeCAMTA4.1* may play major roles in stem development, while *MeCAMTA1*, *MeCAMTA2*, *MeCAMTA4.2*, and *MeCAMTA6* may be primarily involved in root growth and function, highlighting their potential roles in vegetative growth regulation.

### 2.8. Expression Analysis of MeCAMTAs Under Different Treatments

To investigate the expression patterns of *MeCAMTAs* under different treatments, we performed qRT-PCR analysis. Under ABA treatment, the expression patterns of *MeCAMTA1*, *MeCAMTA2*, and *MeCAMTA4.1* showed a trend of first increasing and then decreasing, while *MeCAMTA4.2* and *MeCAMTA6* exhibited an expression pattern of first increasing, then decreasing, and then increasing again. Among them, *MeCAMTA4.1* and *MeCAMTA6* were significantly induced at different time points, whereas *MeCAMTA3* was not induced at any time point (Figure 7A). Under MeJA treatment, the expression trends of *MeCAMTA1*, *MeCAMTA4.1*, and *MeCAMTA4.2* first increased and then decreased; *MeCAMTA2* and *MeCAMTA3* first decreased and then increased; and the expression pattern of *MeCAMTA6* showed an increasing trend. Among them, *MeCAMTA1* and *MeCAMTA4.2* responded significantly at different time points of MeJA treatment (Figure 7B). Under SA treatment, the expression patterns of *MeCAMTA1*, *MeCAMTA4.1*, and *MeCAMTA4.2* showed a trend of first increasing and then decreasing; the expression trends of *MeCAMTA3* and *MeCAMTA6* first decreased, then increased, and then decreased again; and *MeCAMTA2* exhibited an expression trend of first decreasing and then increasing. Among them, *MeCAMTA4.2* responded significantly to SA at different treatment time points (Figure 7C). These results indicate that most members of *MeCAMTAs* are widely involved in the responses to ABA, MeJA, and SA hormone signals, especially *MeCAMTA4.1* and *MeCAMTA4.2*, which may play important roles in the response of cassava to environmental stresses.

Under drought treatment, the expression trends of *MeCAMTA1*, *MeCAMTA3*, and *MeCAMTA6* generally showed a trend of first increasing and then decreasing; the expressions of *MeCAMTA4.1* and *MeCAMTA4.2* showed a downward trend, while the expression pattern of *MeCAMTA2* showed a trend of first decreasing and then increasing. It is worth noting that the expression level of *MeCAMTA1* was significantly upregulated at different time points under drought treatment, while the expressions of *MeCAMTA2*, *MeCAMTA4.1*, and *MeCAMTA4.2* were inhibited (Figure 8A). Under Mannitol treatment, the expression trends of *MeCAMTA2*, *MeCAMTA3*, and *MeCAMTA6* generally showed a trend of first decreasing, then increasing, and then decreasing again; the expressions of *MeCAMTA1*, *MeCAMTA4.1*, and *MeCAMTA4.2* generally showed an “increasing–decreasing–increasing” trend. Among them, the expressions of *MeCAMTA4.1* and *MeCAMTA4.2* were significantly upregulated at all three different time points, while the expression of *MeCAMTA2* was inhibited (Figure 8B). Under PEG treatment, the expression patterns of all *MeCAMTAs* showed a trend of first decreasing, then increasing, and then decreasing again. Among them, the expression of *MeCAMTA4.1* was significantly upregulated at 3 h, while the expressions of other members were all inhibited (Figure 8C). These results indicate that all members of *MeCAMTA* respond to drought, mannitol, and PEG. Among them, *MeCAMTA1* may be a key gene involved in drought stress, while *MeCAMTA4.1* and *MeCAMTA4.2* may play important roles in osmotic stress and oxidative stress in cassava. In addition, through the transcriptome analysis of cassava drought treatment, it was found that the qRT-PCR trends of most members were consistent with those of the transcriptome, but the trends of some members, such as *MeCAMTA4.1* and *MeCAMTA4.2,* were opposite, which might be due to different treatment times and different varieties Appendix A.

After 12 h of NaCl treatment, the expression levels of all members of the *MeCAMTA* family significantly increased, with *MeCAMTA4.2* showing the highest expression level, which was upregulated by nearly 10-fold (Figure 9A). Within 0–12 h of cold treatment, the expression levels of *MeCAMTA3* and *MeCAMTA4.1* significantly decreased, while *MeCAMTA2* and *MeCAMTA6* were significantly upregulated between 0.5 and 3 h of treatment (Figure 9B). During 1–6 h of H_2_O_2_ treatment, the expression levels of *MeCAMTA1* and *MeCAMTA4.1* genes were upregulated, and the expression level of *MeCAMTA4.1* gene was increased by approximately 2.5-fold at 3 h (Figure 9C). The above results indicate that all *MeCAMTAs* members respond to NaCl, low temperatures, and H_2_O_2_ treatments. It is worth noting that *MeCAMTA1*, *MeCAMTA4.1*, and *MeCAMTA4.2* may play important roles in salinity stress and oxidative stress regulation of cassava, while *MeCAMTA2* and *MeCAMTA6* may be the key genes for cassava to respond to low-temperature stress.

### 2.9. Phenotype and Expression Patterns of MeCAMTA Genes in Cassava Under Drought Stress

To identify cassava germplasms with contrasting drought responses, we first performed a preliminary drought-tolerance evaluation under controlled drought stress. From this screen, we selected two representative lines showing opposite phenotypes: a drought-tolerant line (SC11) and a drought-sensitive line (27-4). Subsequently, more detailed phenotypic analyses confirmed that, under drought stress, the sensitive line displayed markedly greater leaf wilting than the tolerant line (Figure 10A). To investigate whether CAMTA family members are associated with these contrasting responses, we measured *MeCAMTA* expression in leaves of the two lines by qRT-PCR. In the drought-tolerant line SC11, four *MeCAMTA* genes (*MeCAMTA1*, *MeCAMTA4.1*, *MeCAMTA4.2*, and *MeCAMTA6*) were significantly induced by drought (Figure 10B). In contrast, in the drought-sensitive line 27-4, the expression of all surveyed *MeCAMTA* genes was significantly repressed under the same conditions (Figure 10C). In conclusion, *MeCAMTA1*, *MeCAMTA4.1*, *MeCAMTA4.2*, and *MeCAMTA6* show opposite expression patterns in two different germplasms with drought responses, indicating that *MeCAMTA1*, *MeCAMTA4.1*, *MeCAMTA4.2*, and *MeCAMTA6* may have important biological functions in cassava drought stress.

## 3. Discussion

Currently, *CAMTA* genes have been widely identified in various plants. *CAMTA* was first discovered in tobacco [24] and subsequently identified in *Arabidopsis thaliana* [25], *Oryza sativa* [26], *Solanum lycopersicum* [27], *Vitis vinifera* [28], *Zea mays* [29], and *Avena sativa* [6]. Abiotic stress significantly impacts plant growth, development, and yield [6]. Studies have shown that *CAMTA* genes play important roles in plant biotic and abiotic stress responses [21,22,23,30]. Research on *CAMTA* in cassava’s stress resistance remains limited, and the identification of *MeCAMTAs* is of great significance for improving cassava’s stress tolerance.

This study identified six *CAMTA* members from the first T2T genome of cassava (Table 1), the same number as in *Arabidopsis thaliana* [25], while the number of *CAMTAs* in wheat reaches 15 [31]. It is inferred that this may be due to gene loss during evolution or differences arising from *Arabidopsis thaliana* and cassava being diploids, while wheat is a hexaploid. Subcellular in silico localization analysis indicates that all *MeCAMTA* members are localized in the nucleus. Meanwhile, studies in other species also support this finding, such as teak *TgCAMTAs*, which are similarly localized in the nucleus [32]. This aligns with their functional characteristics as *CAMTA* transcription factors.

Phylogenetic and homologous analyses reveal that the *MeCAMTAs* family can be divided into three distinct branches (Figure 1). Notably, each branch includes cassava, *Oryza sativa,* and *Arabidopsis thaliana* proteins, with cassava *CAMTAs* showing similar homology to both the dicot *Arabidopsis thaliana* and the monocot *Oryza sativa*. The results indicate that *CAMTA* is relatively conserved during plant evolution (Figure 2B). Analysis of conserved motifs and gene structures revealed that *MeCAMTAs* members contain 12 to 13 exons and 11 to 12 introns, all harboring 10 conserved motifs, suggesting that *MeCAMTAs* share similar gene structures and identical conserved motifs (Figure 3B,D). Additionally, except for *MeCAMTA2,* which lacks one IQ domain, all other members possess CG-1, TIG, ANK (Ankyrin repeat), and IQ domains (Figure 3C). These domain composition features demonstrate that *CAMTAs* constitute a highly conserved gene family evolutionarily.

Cis-acting elements in the promoter region can reflect the expression regulation of related genes [33], thereby modulating plant growth, development, and environmental adaptability. In this process, transcription factors play a central role, among which *CAMTAs* regulate gene expression by specifically recognizing and binding to cis-acting elements in the promoter regions of target genes. The promoter regions of the cassava *CAMTAs* gene family contain various cis-acting elements related to abiotic stress, hormone response, and plant growth and development, such as LTR response elements, ARE response elements, CCAAT-box response elements, TCA-element response elements, ABRE response elements, CGTCA-motif response elements, GARE-motif response elements, TGA-element response elements, CAT-box response elements, GCN4_motif response elements, and O_2_-site response elements (Figure 4B).

Meanwhile, studies have shown that the *AtCAMTA1* mutant exhibits significantly inhibited root growth and development, increased sensitivity, and a markedly reduced survival rate under drought conditions, forming a sharp contrast with wild-type plants. These phenotypic differences suggest that *AtCAMTA1* may participate in plant drought stress response by activating the ABA signaling pathway [21]. Additionally, *AtCAMTA6* may regulate salinity stress tolerance during seed germination through the ABA signaling pathway [34]. Protein–protein interaction network prediction analysis revealed that MeCAMTA1 and MeCAMTA3 interact with ANN1/Annexin. It has been reported that under high-temperature and drought stress, *OsANN1* enhances plant stress tolerance by mediating antioxidant enzymes [35].

Tissue expression pattern analysis showed that *MeCAMTAs* are expressed in leaf, flower, root, seed, fruit, stem, and root tuber (Figure 6), but primarily in root and stem. This suggests that the *MeCAMTA* family may play an important role in plant vegetative growth. Studies have also shown that the expression levels of all members of the *CmoCAMTAs* and *CmaCAMTAs* in the Cucurbitaceae family are higher in roots than in other tissues [36]. To investigate the expression patterns of *MeCAMTAs* under abiotic stress, cassava was analyzed under different treatments (MeJA, PEG, H_2_O_2_, ABA, SA, cold, NaCl, mannitol, and drought). The results indicated that the expression levels of *MeCAMTA1*, *MeCAMTA3*, and *MeCAMTA6* were significantly upregulated under drought treatment (Figure 8A). Notably, similar results were observed in wheat, where the expression of *TaCAMTA1b-B.1* significantly increased after drought stress [37]. Under SA treatment, the expression levels of the other five members, except *MeCAMTA3*, were significantly upregulated (Figure 7C). In citrus, eight *CAMTA* members showed upregulated expression after SA treatment [38]. In addition, under NaCl treatment, the expression levels of all *MeCAMTA* members were significantly upregulated (Figure 9A). Studies have also shown that under salinity stress, the expression of *FaCAMTA3* in strawberries was significantly upregulated [39]. In addition, under drought stress, *MeCAMTAs* exhibit differential gene expression patterns between drought-tolerant and drought-sensitive germplasms. Among them, most *MeCAMTA* genes are significantly upregulated in drought-tolerant germplasms, while the expression patterns of all *MeCAMTAs* show a downregulated expression pattern in drought-sensitive germplasms (Figure 10). These results suggest that *MeCAMTAs* may play an important role in drought stress.

In summary, the *MeCAMTA* gene family plays a critical regulatory role in abiotic stress responses and hormone signaling in cassava, but its biological functions and molecular mechanisms still need further investigation.

## 4. Materials and Methods

### 4.1. Identification of Cassava CAMTA Gene Family Members and Analysis of Their Physicochemical Properties

The T2T genome and annotation documentation of cassava (2*n* = 36) cultivar ‘Xinxuan 048’ (XX048) were obtained from the National Genomics Data Center of China (https://ngdc.cncb.ac.cn/gwh/search/advanced/, accessed on 3 March 2025, accession number: PRJCA016162) [40]. The genome files and annotation documentation of *Arabidopsis thaliana* (TAIR10), *Solanum tuberosum*, and *Oryza sativa* (v7.0) were retrieved from the Phytozome v13 [41] database (https://phytozome-next.jgi.doe.gov/, accessed on 3 March 2025). The genome of *Ricinus communis* was derived from EupDB (http://eupdb.liu-lab.com/keywords_search/, accessed on 20 November 2025) [42]. The CAMTA protein sequences of *Arabidopsis thaliana* were derived from TAIR (https://www.arabidopsis.org/, accessed on 3 March 2025). Using TBtools v2.114 [43], BLAST search was performed on the cassava T2T protein sequences with all CAMTA protein sequences of *Arabidopsis thaliana*, setting the E-value threshold at 1 × 10^−5^. To enhance the accuracy of the appraisal candidate genes, the *Arabidopsis thaliana* CAMTA Protein sequences were submitted to the Pfam [44] database (https://pfam.xfam.org/, accessed on 3 March 2025) to obtain CAMTA domains including CG-1 (PF03859), TIG (PF01833), ANK (ankyrin repeat sequence) (PF12796), and IQ (PF00612) [45], and a hidden Markov model (HMM) search was conducted on the cassava T2T protein sequences. By comparing the results of BLAST and HMM, the candidate protein sequences were finally obtained. Subsequently, the obtained candidate protein sequences were submitted to NCBI-CDD [46] (https://www.ncbi.nlm.nih.gov/cdd, accessed on 5 March 2025) for validation, and the cassava CAMTA Protein sequences were ultimately obtained. The physicochemical properties of CAMTA proteins were predicted using the ExPASy [47] online website (https://web.expasy.org/protparam/, accessed on 5 March 2025). Subcellular localization prediction was performed using Cell-PLoc 2.0 [48] (http://www.csbio.sjtu.edu.cn/bioinf/Cell-PLoc-2/, accessed on 5 March 2025).

### 4.2. Phylogenetic Analysis of MeCAMTAs

Multiple sequence alignment of CAMTA protein sequences from cassava, *Arabidopsis thaliana*, and rice was performed using the ClustalW tool in MEGA11 [49] software with other parameters set to default. Subsequently, a phylogenetic tree was constructed using the Neighbor-joining (NJ) method in MEGA11 software, with Bootstrap validation parameters set to 1000 replicates and other parameters remaining at default values. The phylogenetic tree was beautified using iTOL v7 [50] (https://itol.embl.de/, accessed on 6 March 2025).

### 4.3. Chromosome Localization and Homology Analysis

The chromosomal positions of cassava *CAMTA* genes were visualized using TBtools v2.114. We analyzed the homology of *CAMTA* gene families among cassava, *Arabidopsis thaliana*, *Solanum tuberosum*, and *Oryza sativa*, as well as within the cassava *CAMTA* gene family itself, using the MCScanX [51] tool in TBtools v2.114, and visualized the results with TBtools v2.114.

### 4.4. Conserved Motifs, Functional Domains, and Gene Structure Analysis

The conserved motifs of *MeCAMTA* were identified using the MEME V 5.5.9 [52] online website (https://meme-suite.org/meme/, accessed on 10 March 2025), with the number of motifs set to 10 and other parameters kept at default settings. The structural domain files of the *MeCAMTA* gene family were predicted using the NCBI-CDD v3.20 online tool (https://www.ncbi.nlm.nih.gov/cdd, accessed on 10 March 2025). The gene structures were then extracted from the GFF3 files using TBtools v2.114 software, and the GFF3 files of the six genes are listed in the Appendix A. Finally, the conserved motifs, functional domains, and gene structures were visualized using TBtools v2.114 software.

### 4.5. Promoter Cis-Acting Element Analysis

To analyze the cis-acting elements in the promoter regions, the 1.5 kb sequences upstream of the transcription start sites in the cassava genome were extracted using TBtools v2.114 software. The cis-acting elements of the *MeCAMTA* gene promoters were then analyzed using the PlantCARE [53] online tool (https://bioinformatics.psb.ugent.be/webtools/plantcare/html/, accessed on 15 March 2025) and the results were visualized using TBtools v2.114 software.

### 4.6. Protein–Protein Interaction Network Prediction

The String [54] online tool (https://string-db.org/, accessed on 20 March 2025) was used to predict potential protein–protein interactions of MeCAMTA, and the results were imported into Cytoscape v3.10.3 [55] for visualization refinement.

### 4.7. Plant Materials and Stress Treatments

There are three cassava varieties for experimental materials, including SC124, SC11, and 27-4. Among them, SC11 are drought-tolerant germplasms and 27-4 are drought-sensitive germplasms (all experiments were performed at Hainan University). Stem segments of approximately 3–4 cm in length were planted in plastic pots (16 cm in diameter × 14 cm in height) containing mixed soil (fine sand–vermiculite–nutrient soil = 1:1:1) and cultivated in a constant temperature and humidity greenhouse (growth conditions: light/dark = 16 h/8 h, 30 °C, 70% humidity) for 50 d for subsequent experiments.

The experimental materials were subjected to nine treatments representing hormone stimulation and various abiotic stresses. Hormone treatments included 100 μM MeJA, 100 μM ABA, and 5 mM SA. Abiotic stress treatments included 300 mM PEG and 300 mM mannitol (drought/osmotic stress), 200 mM NaCl (salinity stress), 10 mM H_2_O_2_ (oxidative stress), cold stress, and drought treatment. In addition, a 7-day drought stress assay was performed on drought-tolerant and drought-sensitive germplasms. After each treatment, cassava leaves were collected at different time points, rapidly frozen in liquid nitrogen, and stored at −80 °C until use. For plants grown under normal conditions, samples were collected from multiple tissues, including leaves, flowers, roots, seeds, fruits, and tubers, followed by immediate freezing in liquid nitrogen and storage at −80 °C. All treatments were conducted with three biological replicates.

### 4.8. RNA Extraction, cDNA Synthesis, and qRT-PCR Analysis

Total RNA was extracted from cassava using the Plant Total RNA Extraction Kit (Tiangen Biotech Co., Ltd., Beijing, China). Two micrograms of RNA were reverse-transcribed into cDNA using the Evo M-MLV Reverse Transcription Premix Kit (Accurate Biology Co., Ltd., Changsha, China). qRT-PCR was then performed using the Bio-Rad CFX96 system v2.3 (Bio-Rad, Singapore) and SYBR Green Pro Taq HS Premix qPCR Kit II (Accurate Biology Co., Ltd., Changsha, China). Each sample in the experiment was subjected to 3 biological replicates and 3 technical replicates to ensure the stability of the results. *MeACTIN* was used as the internal reference gene, and the relative expression level of the *MeCAMTA* gene was calculated using the 2^−ΔΔCT^ method [56]. GraphPad Prism 9.5 was employed for statistical analysis of significant differences and to generate visual charts. The primers used in qRT-PCR are listed in Appendix A. In addition, the RNA-seq data of cassava ‘KU50’ and ‘xx048’ under drought treatment was downloaded through the National Genomics Data Center of China (https://ngdc.cncb.ac.cn/gwh/search/advanced/, accessed on 20 November 2025, accession number: PRJNA385393).

## 5. Conclusions

This study identified six *MeCAMTA* family members based on the cassava T2T genome, distributed across four chromosomes. Systematic characterization of this family was accomplished by integrating analyses of gene structure, subcellular localization, conserved motifs, phylogenetic relationships, interaction networks, homology, and cis-acting elements. Tissue expression patterns revealed that *MeCAMTAs* exhibit significant organ specificity and are prominently expressed in roots and stems. qRT-PCR assays demonstrated that all *MeCAMTA* members could be significantly induced by at least one stress condition under MeJA, PEG, H_2_O_2_, ABA, SA, cold, NaCl, mannitol, and drought stress. In summary, the *MeCAMTA* family may play a key role in the abiotic stress and hormone responses of cassava. This study lays the foundation for further elucidating the gene function of *MeCAMTA* in abiotic stress.

## Figures and Tables

**Figure 1 plants-14-03743-f001:**
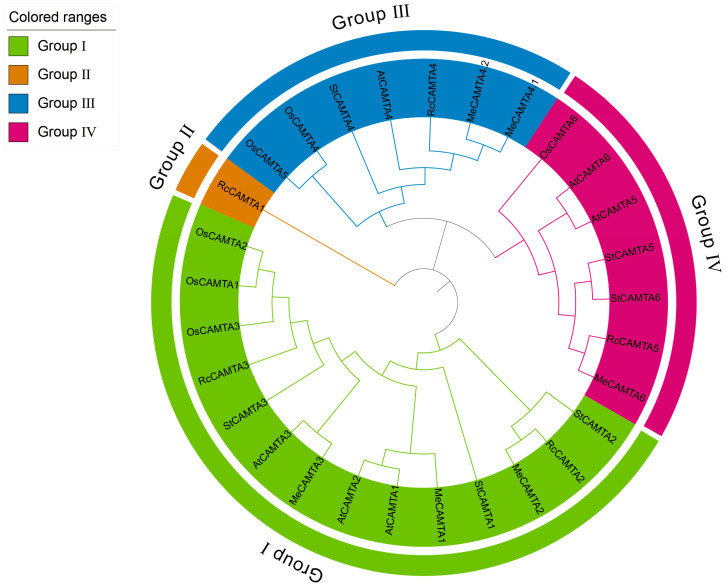
Phylogenetic tree of *CAMTAs* in *Manihot esculenta*, *Arabidopsis thaliana*, *Ricinus communis*, *Solanum tuberosum*, and *Oryza sativa*. The tree was constructed using the Neighbor-joining (NJ) method with 1000 replicates.

**Figure 2 plants-14-03743-f002:**
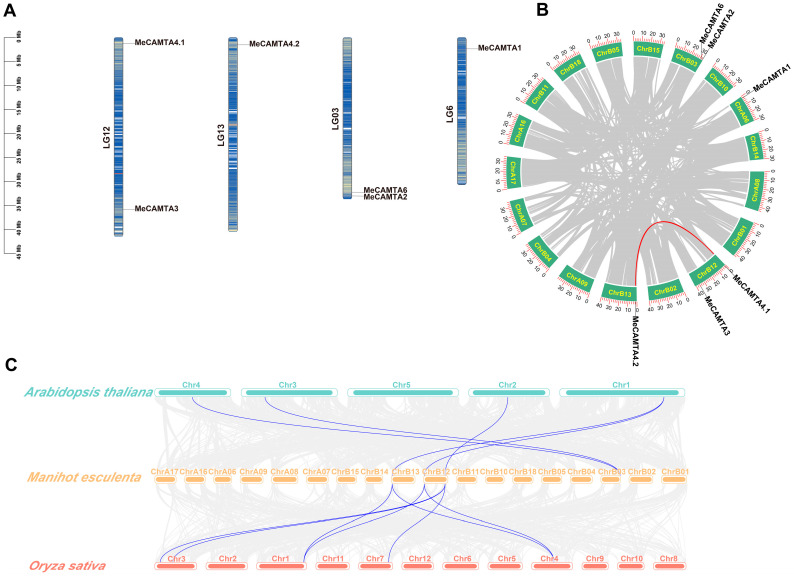
Chromosomal distribution and synteny analysis of *MeCAMTAs.* (**A**) Chromosomal localization of *MeCAMTAs*. The scale bar on the left is in megabases (Mb). Gene density on chromosomes is displayed as a heatmap, with darker regions indicating higher gene density. (**B**) Intra-species synteny analysis in cassava. Red lines indicate segmentally replicated gene pairs. (**C**) Synteny analysis among *Manihot esculenta*, *Arabidopsis thaliana*, and *Oryza sativa*. Gray lines represent collinear blocks between the genomes of *Manihot esculenta*, *Arabidopsis thaliana*, and *Oryza sativa*. Blue lines indicate homologous gene pairs among the genomes of *Manihot esculenta*, *Arabidopsis thaliana*, and *Oryza sativa*.

**Figure 3 plants-14-03743-f003:**
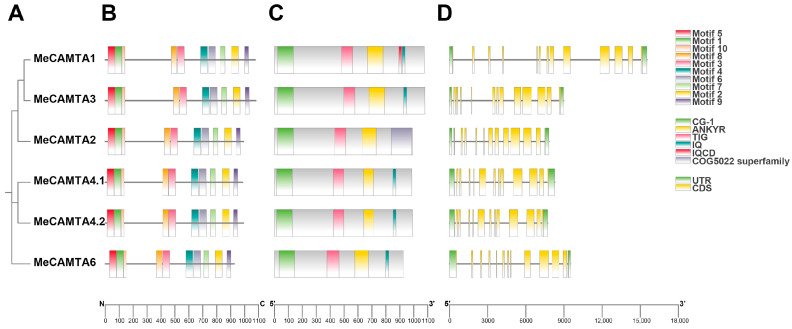
Phylogenetic tree, conserved motifs, conserved domains, and gene structure of *MeCAMTAs*. (**A**) Phylogenetic tree of *MeCAMTAs*. The phylogenetic tree was constructed using the protein sequences of *MeCAMTAs* by the Neighbor-joining (NJ) method with 1000 replicates. (**B**) Conserved motifs of *MeCAMTAs*. Ten conserved motifs are represented by boxes of different colors. (**C**) Conserved domains of *MeCAMTAs*. Conserved domains are represented by boxes of different colors. (**D**) Gene structure of *MeCAMTAs*. Exon–intron structure: yellow boxes represent exons (CDS), black lines represent introns, and green boxes represent untranslated regions (UTR).

**Figure 4 plants-14-03743-f004:**
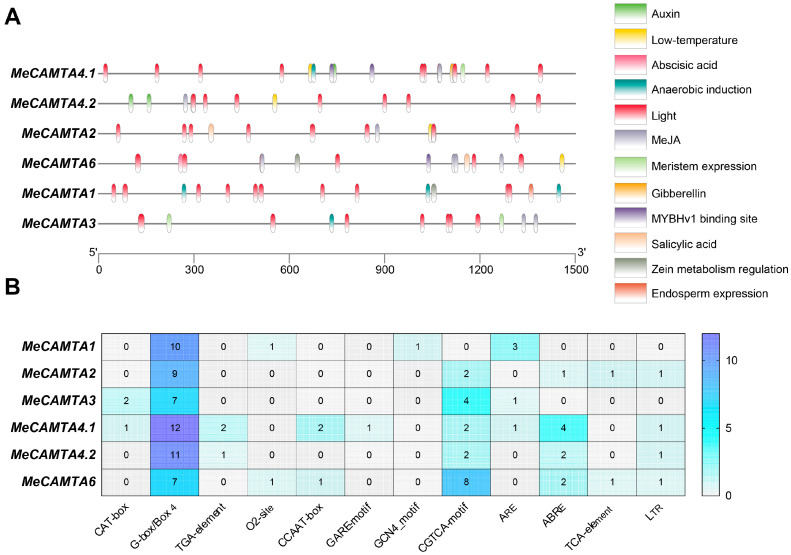
Analysis of cis-acting elements in the promoter region of *MeCAMTAs*. (**A**) Cis-acting elements in the promoter region of the *MeCAMTAs*. The promoter region contains 12 types of cis-acting elements, each represented by a differently colored box. The scale bar below indicates the positions of the cis-acting elements within the promoter region. (**B**) The number of cis-acting elements in the promoter region of *MeCAMTAs*. The numbers in the grid represent the number of cis-acting elements.

**Figure 5 plants-14-03743-f005:**
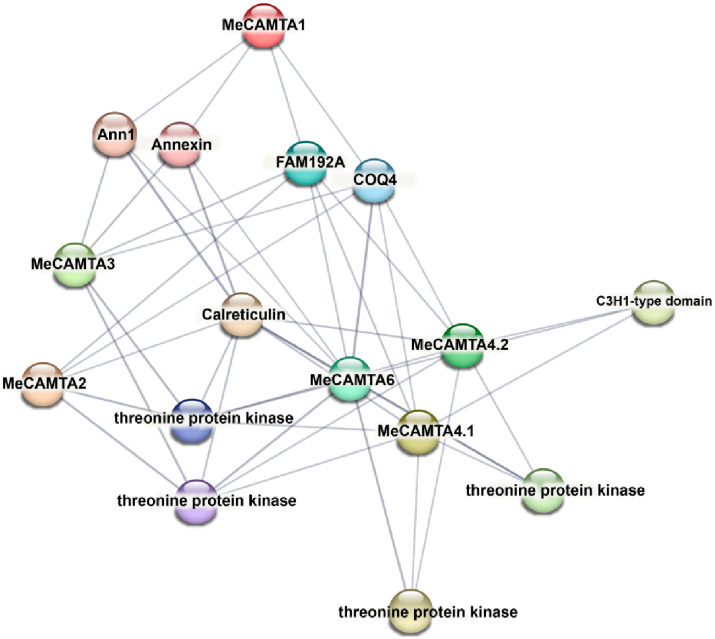
Prediction analysis of MeCAMTAs protein–protein interaction (PPI) network.

**Figure 6 plants-14-03743-f006:**
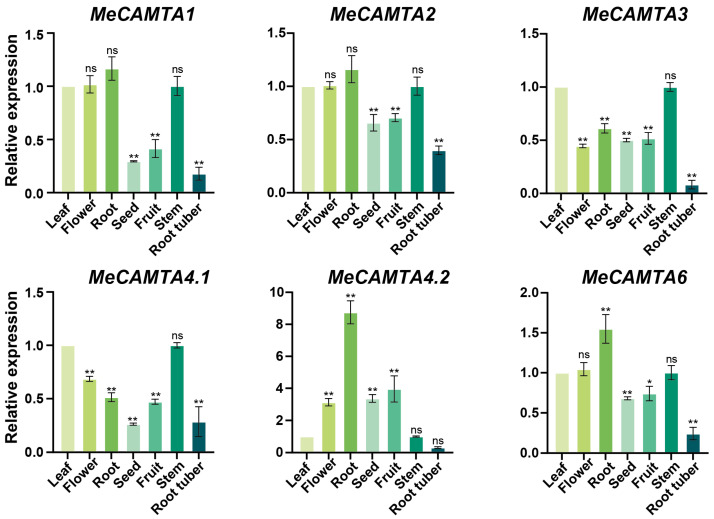
Expression patterns of *MeCAMTAs* in different tissues. The asterisk (*) indicates upregulated or downregulated gene expression levels compared to the control. (* *p* < 0.05, ** *p* < 0.01, and ‘ns’ indicates non-significant differences, one-way ANOVA, *t*-test).

**Figure 7 plants-14-03743-f007:**
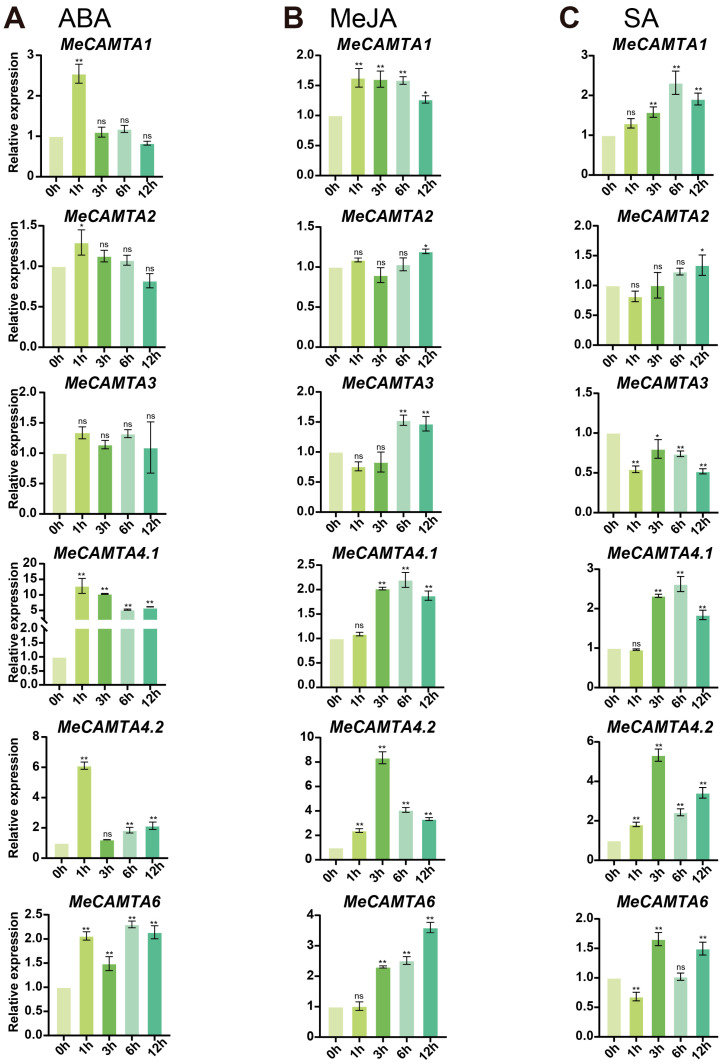
Expression patterns of *MeCAMTAs* under (**A**) ABA, (**B**) MeJA, and (**C**) SA treatments. Asterisks (*) denote upregulated or downregulated gene expression levels compared to the control. * *p* < 0.05, ** *p* < 0.01, and ‘ns’ indicates non-significant differences, one-way ANOVA, *t*-test).

**Figure 8 plants-14-03743-f008:**
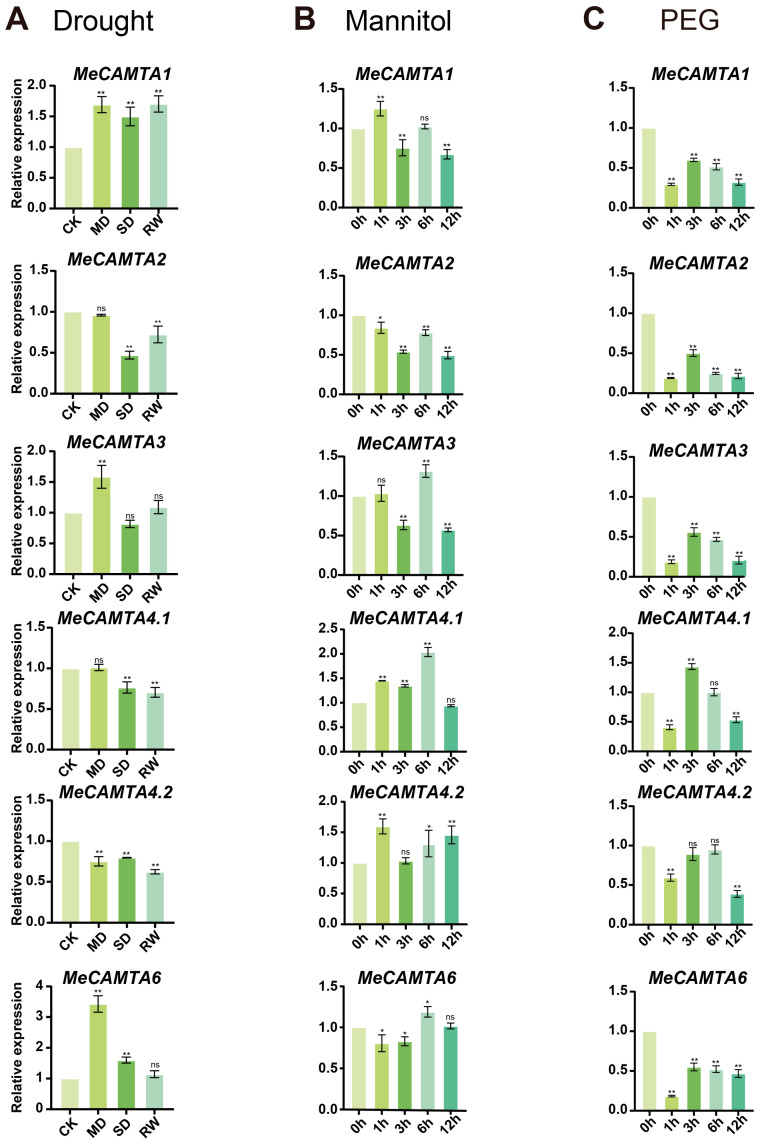
Expression patterns of *MeCAMTAs* under (**A**) drought, (**B**) mannitol, and (**C**) PEG treatments. MD indicates mild drought, SD indicates severe drought, and RW indicates after rewatering. Asterisks (*) indicate upregulated or downregulated gene expression levels compared to the control. (* *p* < 0.05, ** *p* < 0.01, and ‘ns’ indicates non-significant differences, one-way ANOVA, *t*-test).

**Figure 9 plants-14-03743-f009:**
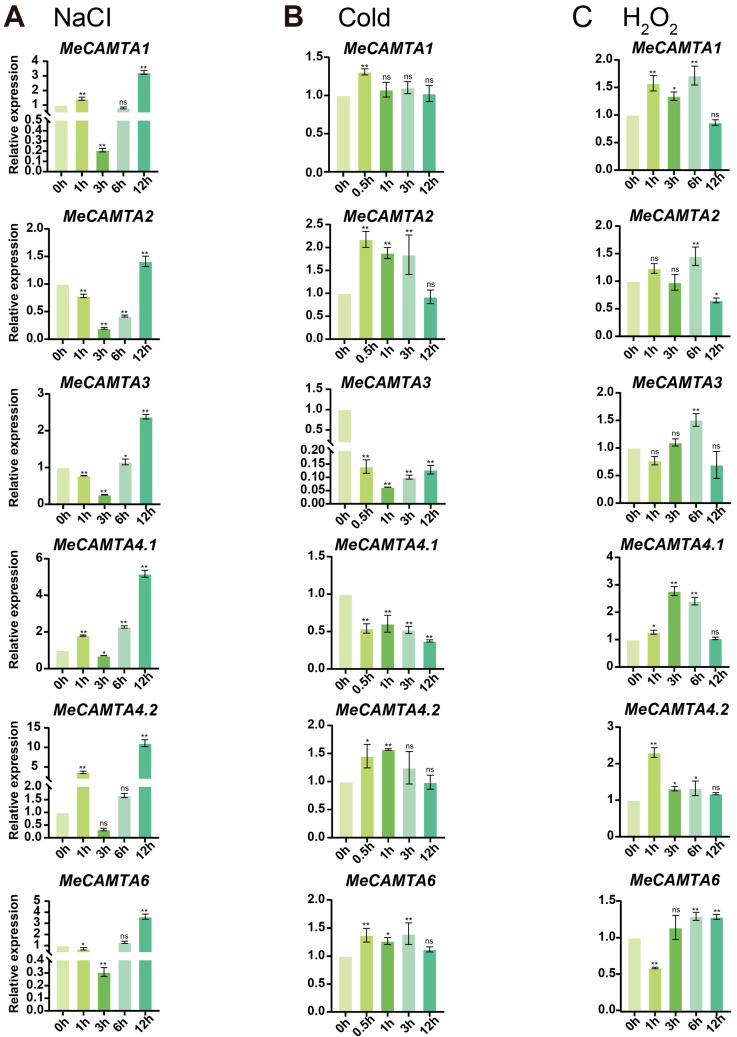
Expression patterns of *MeCAMTAs* under (**A**) NaCl, (**B**) cold, and (**C**) H_2_O_2_ treatments. The asterisk (*) indicates upregulated or downregulated gene expression levels compared to the control. (* *p* < 0.05, ** *p* < 0.01, and ‘ns’ indicates non-significant differences, one-way ANOVA, *t*-test).

**Figure 10 plants-14-03743-f010:**
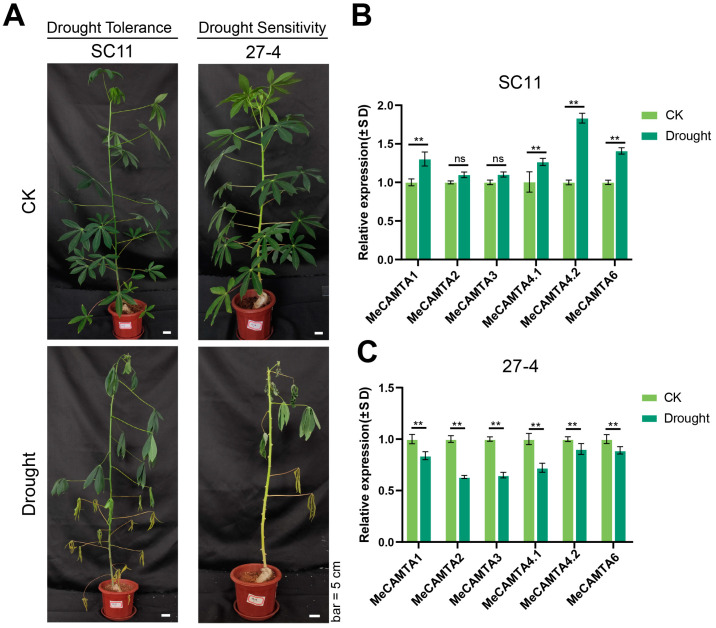
Analysis of cassava phenotypes and expression patterns of *MeCAMTAs* under drought Treatment. (**A**) Phenotypic analysis of drought-tolerant germplasms (SC11) and drought-sensitive germplasms (27-4) under drought stress treatment. CK: control group. Drought: Drought treatment group. (**B**,**C**) Expression patterns of *MeCAMTAs* in drought-tolerant germplasms (SC11) and drought-sensitive germplasms (27-4) under drought treatment. The asterisk (*) indicates upregulated or downregulated gene expression levels compared to the control. (** *p* < 0.01, and ‘ns’ indicates non-significant differences, one-way ANOVA, *t*-test).

**Table 1 plants-14-03743-t001:** Physicochemical properties of *CAMTA* gene family in Cassava.

Gene Name	Gene ID	Length	MW	pI	GRAVY	Instability Index	Subcellular Localization
*MeCAMTA1*	DescChrA06G00699830.1	1075	120,146.68	5.51	−0.499	52.60	Nucleus
*MeCAMTA2*	DescChrB03G00619160.1	991	111,272.25	6.99	−0.533	40.13	Nucleus
*MeCAMTA4.1*	DescChrB12G00091490.1	985	110,248.97	5.49	−0.571	45.02	Nucleus
*MeCAMTA3*	DescChrB12G00106750.1	1079	120,796.47	5.94	−0.540	40.94	Nucleus
*MeCAMTA4.2*	DescChrB13G00136590.1	991	111,101.60	5.62	−0.510	47.43	Nucleus
*MeCAMTA6*	DescChrB03G00618330.1	925	104,769.73	6.67	−0.429	38.65	Nucleus

## Data Availability

The original contributions presented in this study are included in the article/Appendix A. Further inquiries can be directed to the corresponding author(s).

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
