# Peer review of "Genome-Wide Identification and Expression Analysis of CAMTA Genes in Cassava Under Abiotic Stresses"

_plants, 2025, doi:10.3390/plants14243743_

Round 1
Reviewer 1 Report
Comments and Suggestions for Authors
What needs to be improved in this work is the quality of the figures, especially the descriptions on the figures. Most of them are illegible and, for example, make it impossible to verify the significance of statistical differences.
Several sentences need to be corrected, including those that seem illogical to me and concern the number of introns.
I have noted the remaining comments in the attached file.

Author Response
- Summary
Thank you very much for taking the time to review our manuscript. Please find our detailed responses below, and the corresponding revisions and corrections have been marked using 'Track Changes' in the resubmitted files.
2.Point-by-point response to Comments and Suggestions for Authors
Reviewers' comments:
Reviewer #1
Comments 1. What needs to be improved in this work is the quality of the figures, especially the descriptions on the figures. Most of them are illegible and, for example, make it impossible to verify the significance of statistical differences.
Response 1: Thank you for your comments. I have revised all the pictures in the manuscript according to your suggestion, replacing them with clear ones so that the captions can be clearly seen.
Comments 2. Several sentences need to be corrected, including those that seem illogical to me and concern the number of introns.
Response 2: Thank you for your comments. I have revised the manuscript according to your suggestions. The corresponding changes can be seen in lines 153-156.
Comments 3. Formatting issue: Incorrect capitalization of names in multiple sentences throughout the article.
Response 3: Thank you for your comments. I have revised the manuscript according to your suggestions.
Comments 4. Line 95: The phrasing "Group III consisted solely of MeCAMTA6" is unclear.
Response 4: Thank you for your comments. I have revised the manuscript according to your suggestions. The corresponding changes can be seen in line 97.
Comments 5. Line193: "the lowest in root tubers." this is not in contrast!
Response 5: Thank you for your comments. I have revised the manuscript according to your suggestions. The corresponding changes can be seen in lines 208.
Comments 6. Figure captions are excessively lengthy.
Response 6: Thank you for your comments. I have revised the manuscript according to your suggestions. The corresponding changes can be seen in lines 241-244,273-277,309-313.
Comments 6. To my knowledge PEG cause osmotic stress not oxidative one.
Response 6: Thank you for your comments. In this sentence, osmotic stress refers to PEG, and oxidative stress refers to Mannitol. The corresponding content can be found on lines 266.
Comments 7. Inappropriate wording.
Response 7: Thank you for your comments. I have revised the manuscript according to your suggestions. The corresponding changes can be seen in lines 353.
Comments 8. This caption must be corrected.
Response 8: Thank you for your comments. I have revised the manuscript according to your suggestions. The corresponding changes can be seen in lines 535.
Reviewer 2 Report
Comments and Suggestions for Authors
This manuscript provides a timely and important genome-wide analysis of the CAMTA gene family in cassava, utilizing the latest T2T genome assembly. The identification and characterization of six MeCAMTA genes and their expression analysis under various abiotic stresses is a valuable contribution to understanding stress tolerance in this key crop.
The work provides valuable resources for the cassava research community. However, there were so many typos and inappropriate formatting in this manuscript. I recommend the manuscript for publication after the authors address the following major and minor points.
Major Comments:
Comparison with Related Species: Cassava belongs to the Malpighiales order. The manuscript must include a comparative analysis of CAMTA gene numbers, phylogenetic structure, and synteny against at least one or two closely related or agriculturally relevant species outside of Arabidopsis (e.g., rubber tree, castor bean, or another root/tuber crop). This analysis will help determine if the cassava CAMTA family has undergone unique expansion or contraction events relevant to its adaptation as a root crop.
This manuscript lacks evolution analysis of these genes. In line 111, the authors identified a pair of duplicated genes (MeCAMTA4.1/MeCAMTA4.2). I suggest the authors explicitly state whether this refers to segmental duplication, tandem duplication, or whole-genome duplication (WGD) events. The methods should briefly describe how these events were identified (e.g., using synteny analysis). Furthermore, the Ka/Ks ratio should be calculated for this pair of genes to determine the selective pressure (purifying, neutral, or positive) acting on them, which is standard practice for analyzing gene duplication.
The manuscript lacks of analysis of public RNA-seq data. The authors must re-evaluate the qRT-PCR data for consistency with the RNA-seq data and clearly discuss any discrepancies. If the qRT-PCR data is robust, it should override the RNA-seq observations in the conclusion.
All of the figures are ambiguous, the authors should provide a high-resolution picture.
Line 248 to 267, the increased-decreased-increased-decreased is not reasonable due to too few of time points.
Minor Comments:
Typographical and Spacing Errors: Line 55: The capitalization of abiotic stresses is unnecessary. Please use lowercase: "Drought, salinity and cold." Line 61: There are substantial spacing errors throughout the manuscript, including missing spaces (e.g., the space before "After genetic mutation of" in Line 61). A thorough proofread is required.
Ambiguous Sentences (Line 101–103): The sentences within this range are unclear and ambiguous. The authors must rewrite them for precision and clarity.
Reference to Techniques (Line 196, 454, 458): Please standardize the terminology for the gene expression analysis. Use the established abbreviation qRT-PCR consistently, and avoid verbose terms like "real-time quantitative RT-PCR analysis" or "Quantitative Real-Time PCR (qRT-PCR)" after the abbreviation has been defined once.
Justify Error Bars: The authors must explicitly explain the source of the error bars in the Leaf samples (Figure 6) and the 0h control (Figure 7). If 0h is the normalizing control, it should typically not have error bars unless a different △△Ct baseline is used.
Inappropriate abbreviation of ABA, MeJA and SA in Figure 7.
Line 220, ns >0.05* p < 0.05, ** p < 0.01, one-way ANOVA, T-test is incomplete and confusing. The authors should be more attention to the writing.
Line 233, use Figure 8B.
Line 384, March 3, 202?
Structural Placement (Line 435–436): The introductory material describing cassava varieties should be moved to a more appropriate location, either the last paragraph of the Introduction or the beginning of the Results section, where it provides context for the experimental material.
Reference Formatting: Despite previous revisions, the References section still needs better formatting to ensure strict adherence to the journal's guidelines (e.g., checking for consistent journal abbreviations, full author lists, and standardizing all bibliographic elements).
Author Response
- Summary
Thank you very much for taking the time to review our manuscript. Please find our detailed responses below, and the corresponding revisions and corrections have been marked using 'Track Changes' in the resubmitted files.
- Point-by-point response to Comments and Suggestions for Authors
Reviewers' comments:
Reviewer #2
Comments 1. Comparison with Related Species: Cassava belongs to the Malpighiales order. The manuscript must include a comparative analysis of CAMTA gene numbers, phylogenetic structure, and synteny against at least one or two closely related or agriculturally relevant species outside of Arabidopsis (e.g., rubber tree, castor bean, or another root/tuber crop). This analysis will help determine if the cassava CAMTA family has undergone unique expansion or contraction events relevant to its adaptation as a root crop.
Response 1: Thank you for your comments. I have revised the manuscript according to your suggestions. The corresponding changes can be seen in lines 100-101.
Comments 2. This manuscript lacks evolution analysis of these genes. In line 111, the authors identified a pair of duplicated genes (MeCAMTA4.1/MeCAMTA4.2). I suggest the authors explicitly state whether this refers to segmental duplication, tandem duplication, or whole-genome duplication (WGD) events. The methods should briefly describe how these events were identified (e.g., using synteny analysis). Furthermore, the Ka/Ks ratio should be calculated for this pair of genes to determine the selective pressure (purifying, neutral, or positive) acting on them, which is standard practice for analyzing gene duplication.
Response 2: Thank you for your comments. I have revised the manuscript according to your suggestions, DupGen_finder comparative analysis revealed that the gene pair MeCAMTA4.1/MeCAMTA4.2 originated from a whole-genome duplication (WGD) event. Furthermore, the calculated Ka/Ks ratio was less than 1, indicating that this gene pair has undergone purifying selection.
Comments 3. The manuscript lacks of analysis of public RNA-seq data. The authors must re-evaluate the qRT-PCR data for consistency with the RNA-seq data and clearly discuss any discrepancies. If the qRT-PCR data is robust, it should override the RNA-seq observations in the conclusion.
Response 3: Thank you for your comments. I have revised the manuscript according to your suggestions. The corresponding changes can be seen in lines 267-271.
Comments 4. All of the figures are ambiguous; the authors should provide a high-resolution picture.
Response 4: Thank you for your comments. I have revised the manuscript according to your suggestions. I have revised all the pictures in the manuscript according to your suggestion, replacing them with clear ones so that the captions can be clearly seen.
Comments 5. Line 248 to 267, the increased-decreased-increased-decreased is not reasonable due to too few of time points.
Response 5: Thank you for your comments. I have revised the manuscript according to your suggestions. The corresponding changes can be seen in lines 280-287.
Comments 6. Typographical and Spacing Errors: Line 55: The capitalization of abiotic stresses is unnecessary. Please use lowercase: "Drought, salinity and cold." Line 61: There are substantial spacing errors throughout the manuscript, including missing spaces (e.g., the space before "After genetic mutation of" in Line 61). A thorough proofread is required.
Response 6: Thank you for your comments. I have revised the manuscript according to your suggestions. The corresponding changes can be seen in lines 55 and 61.
Comments 7. Ambiguous Sentences (Line 101–103): The sentences within this range are unclear and ambiguous. The authors must rewrite them for precision and clarity.
Response 7: Thank you for your comments. I have revised the manuscript according to your suggestions. The corresponding changes can be seen in lines 106-109.
Comments 8. Reference to Techniques (Line 196, 454, 458): Please standardize the terminology for the gene expression analysis. Use the established abbreviation qRT-PCR consistently, and avoid verbose terms like "real-time quantitative RT-PCR analysis" or "Quantitative Real-Time PCR (qRT-PCR)" after the abbreviation has been defined once.
Response 8: Thank you for your comments. I have revised the manuscript according to your suggestions. The corresponding changes can be seen in lines 220,494,499
Comments 9. Justify Error Bars: The authors must explicitly explain the source of the error bars in the Leaf samples (Figure 6) and the 0h control (Figure 7). If 0h is the normalizing control, it should typically not have error bars unless a different △△Ct baseline is used.
Response 9: Thank you for your comments. I have revised the manuscript according to your suggestions. The corresponding changes can be seen in Figure 6 and Figure 7.
Comments 10. Inappropriate abbreviation of ABA, MeJA and SA in Figure 7.
Response 10: Thank you for your comments. Abscisic Acid (ABA), Methyl Jasmonate (MeJA) and Salicylic Acid (SA).
Comments 11. Line 220, ns >0.05* p < 0.05, ** p < 0.01, one-way ANOVA, T-test is incomplete and confusing. The authors should be more attention to the writing.
Response 11: Thank you for your comments. I have revised the manuscript according to your suggestions. The corresponding changes can be seen in lines 245,279,315.
Comments 12. Line 233, use Figure 8B.
Response 12: Thank you for your comments. I have revised the manuscript according to your suggestions. The corresponding changes can be seen in line 260.
Comments 13. Line 384, March 3, 202?
Response 13: Thank you for your comments. I have revised the manuscript according to your suggestions.
Comments 14. Structural Placement (Line 435–436): The introductory material describing cassava varieties should be moved to a more appropriate location, either the last paragraph of the Introduction or the beginning of the Results section, where it provides context for the experimental material.
Response 14: Thank you for your comments. I have revised the manuscript according to your suggestions.
Comments 15. Reference Formatting: Despite previous revisions, the References section still needs better formatting to ensure strict adherence to the journal's guidelines (e.g., checking for consistent journal abbreviations, full author lists, and standardizing all bibliographic elements).
Response 15: Thank you for your comments. I have revised the manuscript according to your suggestions.
Round 2
Reviewer 2 Report
Comments and Suggestions for Authors
The authors have successfully addressed the majority of my previous concerns, significantly improving the quality of the manuscript. However, several important comments—both major and minor—still need to be addressed before the manuscript can be accepted for publication.
Major Comments:
To allow for full validation of the updated exon/intron structures mentioned in lines 153-160, the complete gene (CDS) and corresponding protein FASTA sequences and gene annotation (GFF file) for all six MeCAMTA genes must be provided as a supplementary file. Currently, I am unable to download or access this necessary sequence information to independently verify the structural updates. And this information is also valuable to other readers.
Minor Comments:
There are still many formatting faults inside this revised manuscript, even in abstract section. Line 19 “tissues,among”, “in stems,while”. The authors must thoroughly review the entire manuscript to ensure correct spacing, punctuation, and consistent use of italics for all gene names and species names.
Line 93 contains a grammatical error: "phylogenetic four?" This phrasing appears to be a newly introduced fault.
In Figure 1, the presentation of the four phylogenetic groups (colored ranges) should be improved. It is recommended that the four colored ranges be arranged sequentially in the legend or on the figure itself to enhance clarity and readability.
The CAMTA genes in Manihot esculenta, Arabidopsis thaliana, Solanum tuberosum, and Oryza sativa demonstrate evolutionary conservation, but asserting that they "originated from a common ancestor" is overly strong and requires more precise language.
Ricinus communis should be italic.
Regarding Line 510, while the text mentions that expression data for cassava accessions ‘KU50’ and ‘xx048’ under drought treatment were downloaded, Figure B3 displays the labels CKU50, SKU50, Cxx048, and Sxx048. I interpret that CKU50 refers to KU50-Control (CK), and SUK50 refers to K50-Salt. If this interpretation is correct, please modify the sample names in Figure B3, or provide a clear legend, to eliminate this ambiguity.
Author Response
- Summary
Thank you very much for taking the time to review our manuscript. Please find our detailed responses below, and the corresponding revisions and corrections have been marked using 'Track Changes' in the resubmitted files.
- Point-by-point response to Comments and Suggestions for Authors
Reviewers' comments:
Reviewer #2
Comments 1. To allow for full validation of the updated exon/intron structures mentioned in lines 153-160, the complete gene (CDS) and corresponding protein FASTA sequences and gene annotation (GFF file) for all six MeCAMTA genes must be provided as a supplementary file. Currently, I am unable to download or access this necessary sequence information to independently verify the structural updates. And this information is also valuable to other readers.
Response 1: Thank you for your comments. I have revised the manuscript according to your suggestions. The corresponding revisions can be found on lines 150-153. The complete gene (CDS) and corresponding protein FASTA sequences and gene annotation (GFF file) can be found in the following Appendix B.
Comments 2. There are still many formatting faults inside this revised manuscript, even in abstract section. Line 19 “tissues,among”, “in stems,while”. The authors must thoroughly review the entire manuscript to ensure correct spacing, punctuation, and consistent use of italics for all gene names and species names.
Response 2: Thank you for your comments. I have revised the manuscript according to your suggestions. The corresponding changes can be seen in line 19.
Comments 3. Line 93 contains a grammatical error: "phylogenetic four?" This phrasing appears to be a newly introduced fault.
Response 3: Thank you for your comments. I have revised the manuscript according to your suggestions. The corresponding changes can be seen in line 92.
Comments 4. In Figure 1, the presentation of the four phylogenetic groups (colored ranges) should be improved. It is recommended that the four colored ranges be arranged sequentially in the legend or on the figure itself to enhance clarity and readability.
Response 4: Thank you for your comments. I have revised the manuscript according to your suggestions. The corresponding changes can be seen in Figure 1.
Comments 5. The CAMTA genes in Manihot esculenta, Arabidopsis thaliana, Solanum tuberosum, and Oryza sativa demonstrate evolutionary conservation, but asserting that they "originated from a common ancestor" is overly strong and requires more precise language.
Response 5: Thank you for your comments. I have revised the manuscript according to your suggestions. The corresponding changes can be seen in lines 129-131.
Comments 6. Ricinus communis should be italic.
Response 6: Thank you for your comments. I have revised the manuscript according to your suggestions. The corresponding changes can be seen in line 423.
Comments 7. Regarding Line 510, while the text mentions that expression data for cassava accessions ‘KU50’ and ‘xx048’ under drought treatment were downloaded, Figure B3 displays the labels CKU50, SKU50, Cxx048, and Sxx048. I interpret that CKU50 refers to KU50-Control (CK), and SUK50 refers to K50-Salt. If this interpretation is correct, please modify the sample names in Figure B3, or provide a clear legend, to eliminate this ambiguity.
Response 7: Thank you for your comments. I have revised the manuscript according to your suggestions. The corresponding changes can be seen in Figure S3.
Round 3
Reviewer 2 Report
Comments and Suggestions for Authors
The authors have significantly improved the quality of the manuscript and the underlying research in response to the collective feedback provided during the review process.
It must be noted that the earlier versions of the manuscript contained a number of errors and omissions, suggesting an initial unfamiliarity with certain established methodologies and a submission that was not fully prepared. This revised manuscript stands as a testament to the value of the peer-review process, demonstrating how the constructive expertise and pro bono contributions of the reviewers have facilitated a substantial enhancement of the research and its presentation.
All major and minor issues have now been sufficiently addressed, leading to a robust and comprehensive paper.
Recommendation: Accept.
Congratulations to the authors on the successful revision of their manuscript.